# Early Detection of Avian Diseases Based on Thermography and Artificial Intelligence

**DOI:** 10.3390/ani13142348

**Published:** 2023-07-19

**Authors:** Mohammad Sadeghi, Ahmad Banakar, Saeid Minaei, Mahdi Orooji, Abdolhamid Shoushtari, Guoming Li

**Affiliations:** 1Biosystems Engineering Department, Tarbiat Modares University, Tehran 14117-13116, Iran; mohammad.sadeghi@modares.ac.ir (M.S.); minaee@modares.ac.ir (S.M.); 2Department of Medical Engineering, Tarbiat Modares University, Tehran 14117-13116, Iran; 3Department of Poultry Disease, Razi Vaccine and Serum Research Institute, Karaj 31976-19751, Iran; hamid1342ir@yahoo.com; 4Department of Poultry Science, Institute for Artificial Intelligence, University of Georgia, Athens, GA 30602, USA

**Keywords:** avian disease, poultry, precision livestock farming, machine learning, thermography

## Abstract

**Simple Summary:**

From an economic point of view, timely information about the flock state is crucial for poultry farmers. When a flock is infected with a disease, if quick and necessary measures are not taken, the disease will spread and affect the whole flock. Artificial intelligence is one of the popular methods in precision livestock farming and is effective in various fields such as weight measurement, feed intake estimation, and disease diagnosis. So far, chicken disease has been diagnosed using sound signal processing and video recordings. This study attempted to develop a new and rapid method of poultry disease diagnosis based on thermography for data collection and artificial intelligence for data analytics. With the proposed method, Avian Influenza and Newcastle Disease can be detected within 24 h after virus infection.

**Abstract:**

Non-invasive measures have a critical role in precision livestock and poultry farming as they can reduce animal stress and provide continuous monitoring. Animal activity can reflect physical and mental states as well as health conditions. If any problems are detected, an early warning will be provided for necessary actions. The objective of this study was to identify avian diseases by using thermal-image processing and machine learning. Four groups of 14-day-old Ross 308 Broilers (20 birds per group) were used. Two groups were infected with one of the following diseases: Newcastle Disease (ND) and Avian Influenza (AI), and the other two were considered control groups. Thermal images were captured every 8 h and processed with MATLAB. After de-noising and removing the background, 23 statistical features were extracted, and the best features were selected using the improved distance evaluation method. Support vector machine (SVM) and artificial neural networks (ANN) were developed as classifiers. Results indicated that the former classifier outperformed the latter for disease classification. The Dempster–Shafer evidence theory was used as the data fusion stage if neither ANN nor SVM detected the diseases with acceptable accuracy. The final SVM-based framework achieved 97.2% and 100% accuracy for classifying AI and ND, respectively, within 24 h after virus infection. The proposed method is an innovative procedure for the timely identification of avian diseases to support early intervention.

## 1. Introduction

A modern broiler house accommodates tens of thousands of animals, and the number is more for layer houses. The high rearing stocking density is favorable for economic profits but increases the risks of bacterial/virus transmission and causing diseases [1]. For instance, the USDA Animal and Plant Health Inspection Service reported that by 18 May 2023, a total of 325 commercial flocks, 511 backyard flocks, and 58.79 million birds were affected by the highly pathogenic Avian Influenza (HPAI) outbreak [2]. The outbreak severely struck the low-profit-margin poultry industry and led to economic loss for producers. Optimal strategies that help ameliorate the threat of emerging diseases in the poultry industry must include rapid and accurate health assessment, so producers can intervene timely to prevent severe outbreaks.

Manual observation methods are not suitable for this task, as most caretakers are not qualified to diagnose health problems, and they need to identify the issue if there is a potential challenge occurring and then call veterinarians for sampling and diagnosing. Furthermore, clinical signs such as diarrhea and nasal discharge, and other complication symptoms can appear in multiple diseases (e.g., Newcastle Disease “ND”, AI, and fowl cholera) [3], making it difficult to pinpoint from visual observation and provide corresponding treatment. Common methods to assess bird health consist of on-site sampling and subsequent disease diagnosis at a laboratory, which can take several days to receive a diagnosis and requires skilled laboratory operators and veterinarians [4]. Delayed disease diagnosis may lead to the disease progressing to the point where the symptoms are widespread and pronounced [5].

Alternative solutions are to use precision livestock and poultry techniques, as they could provide timely and accurate disease diagnosis results [6]. Automated monitoring systems have brought benefits to farm production management and improvement in animal health and welfare [7,8,9]. Among the wide range of techniques, sound recognition and computer vision systems are popular tools to investigate animal health and welfare status and identify animal behavior and internal situation; they are non-invasive for data collection, ensuring continuous animal monitoring without disturbing [10,11,12,13,14]. Animals use vocalization to express conditions such as warning, alarm, nesting, threat, distress, fear, food, privacy, dominance, and time calls [15,16]. Despite being a useful tool, sound recognition requires microphones placed close to animals to collect accurate audio signals for processing. Close placement of the microphones could result in dust accumulation and damage pecked by birds [17,18]. Instead, computer vision systems are typically installed far from animals, resulting in less dust and dirt accumulation and damage to the equipment. Previous studies have investigated computer vision techniques for disease diagnosis. Minna et al. (2018) identified the sick yellow feather chicken based on head features (e.g., eye and comb) in captured images with the support vector machine (SVM) classifier. The final accuracy of classifying healthy and sick birds was 92.5% [19]. Zhuang et al. (2018) developed an early warning algorithm to detect H5N2 AI for broilers based on their outlines and skeleton information gained from images. They also applied an SVM classifier and obtained 99.46% accuracy for disease recognition [20]. Okinda et al. (2019) designed a machine vision system to detect and predict healthy and sick birds infected with Newcastle Disease Virus. The SVM classifier improved with the Radial Basis Function, achieved 97% accuracy for the disease prediction [21]. Akmomolafe and Medeiros (2021) showed that Avian Influenza and Newcastle Disease Virus can be detected using a convolutional neural network classifier with classification accuracy ranging from 95% to 98% [22]. The abovementioned studies utilized features of spatial variations on single images and temporal changes in bird mobility across multiple frames. Another set of computer vision-based disease recognition is based on feces image which can reflect important features for digestive diseases. Wang et al., (2019) classified normal and abnormal birds based on shape, color, water content, and shape and water in the dropping images. They developed deep learning object detection algorithms (R-CNN and YOLO-V3) for the classification and gained the best mean average precision of 93.3% [23].

Although these studies show great potential for combing computer vision systems and machine learning for poultry disease recognition, few focused on the combination of thermography and machine learning. Animals, if infected with diseases, may exhibit irregular body temperature, which could be captured by thermography [24,25]. Therefore, the aim of this study was to identify avian diseases using thermography image processing and machine learning. The ND and AI were used as examples to evaluate system efficacy and accuracy as they are popular and fatal diseases in the poultry industry.

## 2. Materials and Methods

### 2.1. Experiment Setup

The experiments were carried out at Tarbiat Modares University, Tehran, Iran. The ND virus’s molecular-and-pathological characterization was carried out at RAZI Institute, Karaj, Iran. The ND Virulence type was velogenic strains, and ND tropism was viscerotropism. The AI virus type was H9N2. Twenty 14-day-old Ross 308 broilers were raised in 20 experimental pens (1.20 m long × 0.80 m wide for each), and four groups of birds were used, with each kept in a separate room. Two groups of birds were infected with ND and AI, respectively, by eye drops (0.1 cc for each eye). Figure 1a shows the operations of virus infection for a chicken. The rest two were considered control groups. Each group of birds was kept in separate pens in a room meshed with stainless steel wire, allowing all birds to see each other (Figure 1b). Separating birds into independent can help better understand the disease infection progress, as group-housed birds can be cross-contaminated.

The virus infection was verified based on clinical signs, RT-PCR test, and virus isolation from infected tissues. The RT-PCR test was conducted using protocol 2, which was described in [26]. Birds needed feed and water intake during the first 8 h after the lights were turned on, and bird infection was conducted after that to avoid bird stress. A total of seven sections were planned. Section 1 indicates the 8th hour after disease infection, Section 2 indicates the 16th hour after disease infection, …, and Section 7 indicates the 56th hour after disease infection.

Thermal images were captured using a FLIR a65 thermal camera with a resolution of 640 × 512 pixels. The camera has good performance within the ambient temperature of −25 to 135 °C (Figure 2a). The captured images were loaded into FLIR Tools software version 4.1 to remove background (Figure 2b), and then the preprocessed images were further enhanced in MATLAB 2020 (Math works Ins., Natic, MA, USA). No existing packages were used in the following sections, and all steps were coded with the listed formulas. The total number of chickens was 80 (40 birds for ND and control samples and 40 birds for AI and control samples), and 240 thermal images were collected from each section (three images were taken from each bird). The experiment was performed within three consecutive days after virus infection. The emissivity of thermography and distance between the camera and birds were 0.95 and 50 cm, respectively. The close-distance data collection can be achieved by movable robotic systems once available. As ambient temperature and relative humidity could influence the accuracy of thermography, they were measured for each image using a digital temperature humidity meter (Figure 3).

The proposed algorithm framework to identify avian diseases (ND and AI) based on thermal images and machine learning is shown in Figure 4. The thermal images of chickens were captured by FLIR camera and preprocessed in FLIR software and Matlab 2020 software. The preprocessing steps included de-noising, background removal, and image enhancement (erosion and dilation) using Image Region Analyzer and Image Segmenter toolbox. In the data mining step, 23 statistical features were extracted from each image, and the best features were selected. The best features were considered as inputs of the machine learning classifier (SVM and ANN). The outputs of the classifiers were improved by the Dempster–Shafer (D-S) evidence theory, after which the diseases were detected.

Deep learning-based techniques, such as convolutional neural networks, can process thermal images directly without feature extraction but require large number of images to obtain robust performance, which we did not have in this study. Instead, extracting the features from thermal images followed by classical machine learning modeling can help us better understand which features are important to gain accurate performance with small datasets. The machine learning-based methods were also GPU-free (without graphical processing units), which could be economically friendly for poultry producers as well.

### 2.2. Feature Extraction and Selection

The raw data contained relatively little information for classification and were not used directly as input for the classifiers. Therefore statistical features were extracted based on intensity information on thermal images. Table 1 shows the formulas for calculating the 23 statistical features, in which *x*(*n*) is the intensity for data points *n* on a thermal image (*n* = 1, 2, …, *N*) [27].

The 23 features, if all fed into the classifiers, could increase model complexity. So, the improved distance evaluation (IDE) procedure has been used for feature dimensionality reduction. Seven steps shown in Figure 5 should be conducted to execute the IDE. Firstly, the mean distance and variance must be computed for intra-class and inter-class. Then, in the fifth step, the reward factor was computed. The best features included the largest intra-class and the lowest inter-class differences. Based on that, the sixth step was to calculate the difference scores of intra-class and inter-class, which was normalized in the seventh step. Finally, the best features were selected based on an arbitrary threshold [28]. The threshold is determined in Section 3.1.

### 2.3. Artificial Neural Network

The ANNs are biologically-inspired computer programs designed to simulate the way in which a human brain processes information. The ANN included an input layer, a hidden layer, and an output layer (Figure 6). Each layer was connected with neurons, and the number of neurons in the input layer was determined by the number of the best features selected by IDE. The number of classes determined the number of neurons in the output layer, and the number of neurons in the hidden layer was decided by trial and error during model training. Various ANNs have been applied in classification, regression, and modeling [29]. The optimal ANN was used to detect avian diseases in this study.

### 2.4. Support Vector Machine

The SVM is a robust classifier first introduced by Cortes and Vapnic in 1995, building the statistical learning theory [30]. The SVM intends to maximize the margin between the two classes. Separating classes using a hypothetical hyperplane is the main idea. Some hyperplanes include the linear, quadratic, and Gaussian Radial Basic Function (RBF) [31]. Figure 7a shows the optimal margin in a linear hyperplane [32], and Figure 7b shows the RBF hyperplane [33]. The SVM was initially introduced for binary classification (two classes) and then applied to solve multiclass problems. The RBF was selected for the hyperplane in the SVM due to its optimal performance [34].

### 2.5. Dempster–Shafer Evidence Theory

The D-S evidence theory was first proposed by Dempster and further developed and refined by Shafer [35]. The D-S has been used in fault diagnosis [36] and disease diagnosis [27]. This theory investigates aspects connected with uncertainty and lack of knowledge and is favorable for solving real-life problems [37]. In this study, whenever ANN and SVM cannot detect the diseases with acceptable accuracy, the D-S will be used in data fusion stage.

### 2.6. Classifier Evaluation Metrics

The metrics to evaluate the developed models included Sensitivity, Specificity, Training, and Testing accuracy for healthy (“acc_healthy”) and unhealthy (“acc_unhealthy”) birds. True Positive, False Positive, True Negative, and False Negative are defined in Table 2 based on a confusion matrix and were calculated to determine the evaluation metrics. The calculation procedures of sensitivity, specificity, and accuracy are presented in Equations (1)–(6).
(1)$$True\;Positive\;Rate=\frac{TP}{TP+FN}$$
(2)$$False\;Positive\;Rate=\frac{FP}{FP+TN}$$
(3)$$Sensitivity=True\;Positive\;Rate=\frac{TP}{TP+FN}$$
(4)$$Specificity=1-False\;Positive\;Rate=1-\frac{FP}{FP+TN}=\frac{FP+TN-FP}{FP+TN}=\frac{TN}{FP+TN}$$
(5)$$acc\_unhealthy=\frac{TP}{TP+FP}$$
(6)$$acc\_healthy=\frac{TN}{TN+FN}$$
(7)$$Testing\;or\;Trainging\;accuracy=\frac{acc\_unhealthy+acc\_healthy}{2}$$

## 3. Results

### 3.1. Data-Mining Results

In this stage, 23 statistical features were extracted from all raw data collected. The AI infection group and the corresponding control group were defined as Group A; the ND infection group and the corresponding control group were defined as Group B. Each feature was scored by IDE, and the best features, which had the most scores, were selected. The outputs of IDE were used as inputs of the classifiers. Table 3 and Table 4 show the feature scores for Groups A and B, respectively. The best threshold for Group A is 0.7 because there was a significant difference between 0.7 and other lower threshold limits. A closer-to-one value indicates better feature quality for classification. So, all features with scores of 0.7 or higher were favorable for classification for Group A. Therefore, F2 (maximum), F4 (quadratic mean square), F9 (root mean square divided by the mean), F21 (the sixth central moment), and F22 (the fourth central moment divided by the square of the variance), which had the feature score of over 0.7 in all seven sections, were selected to identify the AI disease.

Based on Table 4, the best threshold for Group B was 0.8 because there was a significant difference between 0.8 and other lower threshold limits. The F2 (maximum), F3 (standard deviation), F9 (root mean square divided by the mean), F16 (Skewness), and F17 (kurtosis), which had a feature score of over 0.7 in all seven sections, were selected to identify the ND disease.

The thresholds (0.7 for Group A and 0.8 for Group B) were selected manually based on the analysis of the feature scores. For Group A, the difference between the scores for Section 3 and Section 5 does not look significant (same for Group B, Section 4). But most of the feature scores fall into the set thresholds. We wanted to keep consistent thresholds for all groups.

### 3.2. Classifier Performance

This study included 120 thermal images for each group. Table 5 shows the number of thermal images and data splitting for the ANN and SVM development. The data splitting for ANN was 70% for training, 15% for cross-validation, and 15% for testing. The splitting for SVM was 70% for training and 30% for testing. The classifiers were cross-validated, and the average performance was reported. The performance was also used to determine the model parameters, such as Sigma and *C* for SVM and the number of neurons in the hidden layer for ANN.

#### 3.2.1. ANN Performance

As the outputs of IDE were five features for both groups of AI and ND, the number of neurons in the input layer was five, the number of neurons in the output layer was two (because each group had two classes), and the number of neurons in the hidden layer was determined as eight for AI and seven for ND based on the evaluation metric performance. Table 6 shows the ANN performance to detect the AI for all the times for data collection (sections) with the structure of 5 × 8 × 2. The testing accuracy of ANN to detect AI in Section 1 (the 8th hour after virus infection) was 70.37%, while its validation accuracy was 75.93%. Finally, this accuracy reached 100% in Section 7 (the 56th hour after virus infection). The acceptable accuracy of 92.59% was obtained in Section 4. The Specificity in Section 3 or later was 100% which means the ANN did not diagnose any True Positives (the real sickness) as healthy. The Sensitivity in Section 4 was 85.71%, indicating that the classifier cannot accurately exclude False Negative (wrongly identifying healthy birds as unhealthy).

Table 7 shows the ANN performance to detect the ND for all the times for data collection (sections) with the structure of 5 × 7 × 2. The test accuracy of ANN to detect ND was 68.52% in Section 1 and 100% in Section 5. In all sections, the specificity was higher than sensitivity which means that the main problem of ANN was to misdiagnose the healthy birds as unhealthy.

#### 3.2.2. SVM Performance

The outputs of IDE were used as inputs of SVM. The kernel function of RBF and the solver of Sequential Minimal Optimization and Sigma (σ) of 1 were selected as key parameters for SVM (Table 8). The box constraint was 5 for AI prediction and 10 for ND classification.

Table 9 shows the SVM performance to detect the AI in all sections. The classifier obtained 97.22% testing accuracy in Section 3 (the 24th hour after virus infection). The testing accuracy dropped to 77.78% in Section 4 but rebounded to over 94% after Section 5. In comparison, the ANN achieved 100% testing accuracy in Section 4 for classifying AI. The performance discrepancy may be due to the lack of classification ability for SVM in Section 4, where the D-S evidence theory was introduced to improve AI classification accuracy. The D-S evidence theory is applied in Section 3.3.

Table 10 shows the SVM performance to detect the ND in all sections. In Section 1, the overall testing accuracy was 80.56%, related to the high sensitivity (100.00%). In Section 2, the overall testing accuracy even dropped to 78.77%. Similarly, the ANN had low testing accuracy of 74.07% in identifying ND in Section 2 (Table 7). These all indicated the difficulties in accurate ND diagnosis within the 16 h of the disease infection. Parts of the reasons can be found in feature scores calculated by IDE in Table 3 and Table 4, where the feature scores in Section 2 were relatively lower than those in other stages. Therefore, Section 2 was determined as the time point for introducing the D-S evidence theory to improve ND classification accuracy. The D-S evidence theory is applied in Section 3.3.

In Section 3, the SVM reached 100% testing accuracy. In other sections, the performance of Specificity and Sensitivity fluctuated. These could be attributed to the natural regulation of bird body temperature at different hours of the day. The body temperature was the lowest at 1 am, gradually increased until 8 am, dropped until 12 pm, reached maximum from 12 pm to 6 pm, and then dropped again until 1 am. The data collection started at 8 am in Section 1, at 4 pm in Section 2, and at 12 pm in Section 3. In Section 2, both healthy and unhealthy birds increased their body temperature, causing temperature similarities in the thermal images and subsequent poor classification performance. Meanwhile, the virus may not spread fully inside the bird’s body, making the unhealthy birds indistinguishable from healthy birds.

Since the SVM outperformed the ANN, it was selected for further analysis, and the confusion matrix for classifying the two diseases in all sections is presented in Figure 8. The table contains information such as sensitivity, specificity, and overall accuracy of the SVM in both the training and testing stages. In Section 4, the Sensitivity and Specificity were reported as 85.71 and 72.73%, respectively, and SVM wrongfully classified 6 Flu out of 18 as healthy and 2 healthy out of 18 as Flu. On the other hand, In Section 2, the Sensitivity and Specificity were reported as 72.73 and 85.71 percent, respectively, and SVM wrongly classified 2 ND out of 18 as healthy and 6 healthy out of 18 as ND.

### 3.3. Data Fusion Results

As mentioned earlier, whenever ANN and SVM cannot detect the disease with acceptable accuracy, the D-S was used as the data fusion stage. According to the references reviewed in this study, the performance was acceptable when its value was over 80%. The set acceptable performance varies among studies. In Section 3.2, the classifiers had problems detecting AI in Section 4 and ND in Section 2, where the D-S should be introduced. Table 11 and Table 12 show the results of data fusion to identify AI in Section 4 and ND in Section 2 with Dempster–Shafer evidence theory, respectively.

Based on Table 11, the specificity increased from 72.73% (Table 9) to 100%; and the sensitivity increased from 85.71% (Table 9) to 98.15%; the performance of AI classification via SVM in Section 4 has been improved with D-S evidence theory. Based on Table 13, the Sensitivity increased from 72.73% (Table 10) to 82.90%; the Specificity increased from 85.71% (Table 10) to 96.35%; and the performance of ND classification via SVM in Section 2 has been improved with D-S evidence theory.

## 4. Discussion

In general, the SVM outperformed the ANN in identifying chickens infected with AI with higher sensitivity and testing accuracy. For example, the testing accuracy of AI in Section 1 was 86.11% for SVM (Table 9) and 70.37% for ANN (Table 6). This indicates that the SVM may do a better job in alerting producers if birds were infected with AI within the first 8 h of infection, which helps producers take early intervention and reduce economic loss. SVM did better when the input was the features extracted from raw data. But that does not mean SVM outperforms ANN in any classification tasks. Model comparison and tuning are still required to determine the optimal model for specific tasks.

According to Table 7, the performance of ANN to detect ND reduced after Section 5, but this problem was solved by SVM, as indicated in Table 10. Furthermore, in Section 4, the accuracy of the ANN was very low (Table 7), while the SVM obtained over 90% testing accuracy in this section (Table 10). The low accuracy of ANN was related to the inability of this classifier to distinguish ND in Section 4. Other researchers have also confirmed that the SVM performed better than the ANN in poultry disease classification issues. Okinda et al. (2019) compared SVM and ANN for classifying Newcastle disease, and the RBF-SVM, Cubic_SVM, and ANN had 97.8%, 97.1%, and 96.9%, respectively [18]. This topic has also been confirmed in other fields like mechanical faults diagnosis. Kankar et al. (2011) compared SVM and ANN for classifying ball bearings; the SVM had 73.97% and the ANN had 71.23% [38].

Based on Table 11 and Table 12, the classification performance of AI and ND in Sections 4 and 2 was improved by D-S evidence theory. These results agreed with previous research [39,40]. Banakar et al. (2016) developed an intelligent device for diagnosing avian disease based on vocalization and signal processing. In their study, the accuracy of SVM increased from 83.33% to 91.15% after using the D-S evidence theory [27]. Khazaee et al. (2012) presented data fusion methodology by using ANN and SVM classifiers to distinguish between vibration conditions of planetary gears. They showed that the accuracy of the classifiers increases by more than 14% when using D-S evidence [41].

A summary of the SVM performance to diagnose AI and ND is shown in Table 13, which was based on predicting AI in Section 4 and ND in Section 2 with D-S evidence theory. Based on the proposed method in this study, which was based on thermal images and machine learning, the AI and ND can be detected within 24 h after virus infection (Section 3). The SVM may provide an alarm about the presence of the virus even within the first 8 h (Section 1), but reporting accuracy was below 90%, which may result in False Positives and False Negatives for producers, further degrading the confidence of producers in using the product. The best way was to continuously collect the reporting results and gain more confidence before sending a valid alarm within the first 24 h.

Previous research also investigated avian disease detection via machine learning. Sadeghi et al. (2015) identified and classified the chickens infected with Clostridium Perfringens based on vocalization signals and ANN [42]. Banakar et al. (2016) diagnosed avian diseases using signal processing and SVM, and the system achieved an accuracy of 91.15% in classifying the disease within 48 h after virus infection [27]. Okinda e al. (2019) used machine vision systems to diagnose avian diseases. They infected the chickens with ND and classified the disease with a 97% accuracy on the 4th day after virus infection using RBF-SVM. The parameters of this study were optimized based on the appearance and physical characteristics of the bird body [21]. Overall, our study can provide earlier alarms of disease infection with decent accuracy than the previous studies, indicating the great potential of the combined technique (thermography for data collection and machine learning for data analytics) in this space.

Our dataset was relatively small, with 1680 images in seven sections (240 thermal images per section). However, the disease challenge experiments are typically expensive, and this study serves as the first trial for verifying the possibility of the combined techniques for disease diagnosis. More data should be collected to consolidate the results. We did not expect to develop a viable system that can be directly applied in commercial farms. It is unrealistic to infect birds with avian influenza and Newcastle disease in commercial farms to collect the data for system development, as that can damage producers’ profits and create the risks of disease spreading. Therefore, most of the disease challenge experiments are run cautiously in experimental labs with strict biosecurity control, which could result in small datasets, separated bird housing, and close-distance data collection. These are all future directions for system improvement.

## 5. Conclusions

This study investigated the performance of using thermography and machine learning to classify Avian Influenza and Newcastle Disease for 14-day-old broilers. After a series of optimizations (e.g., parameter tuning and model comparison), the Support Vector Machine with Dempster–Shafer Evidence Theory outperformed the Artificial Neural Networks and successfully classified the two diseases within 24 h after virus infection, with 100% sensitivity, over 94% specificity, and over 97% testing accuracy. It is concluded that thermography combined with machine learning is a useful tool for timely disease prediction, which can be properly utilized to set early alarms and reduce producer economic losses.

## Figures and Tables

**Figure 1 animals-13-02348-f001:**
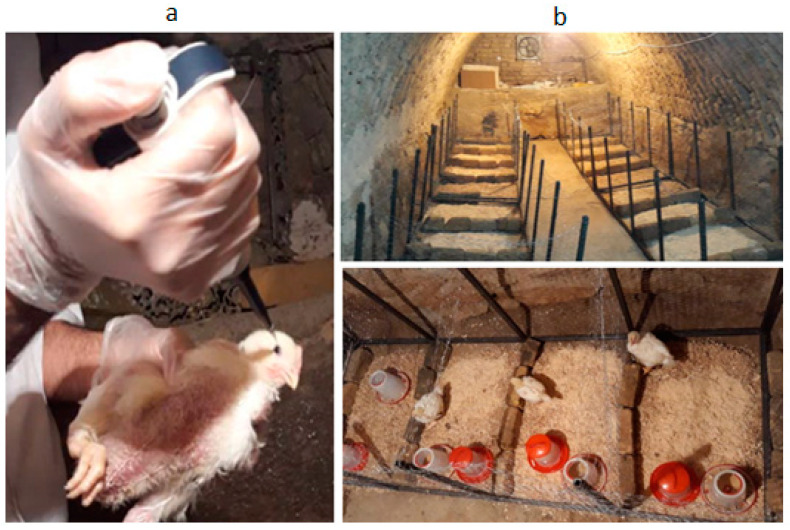
Photos of the experiment: (**a**) Eye drop operation for virus infection; and (**b**) pen scenarios.

**Figure 2 animals-13-02348-f002:**
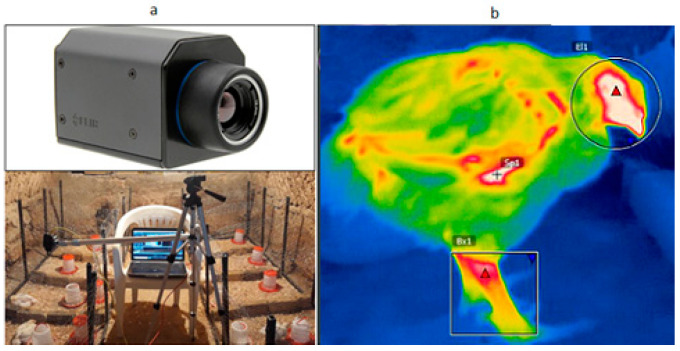
Photos of thermal image collection: (**a**) FLIR a65 camera and camera setup; and (**b**) a thermal image with a bird only.

**Figure 3 animals-13-02348-f003:**
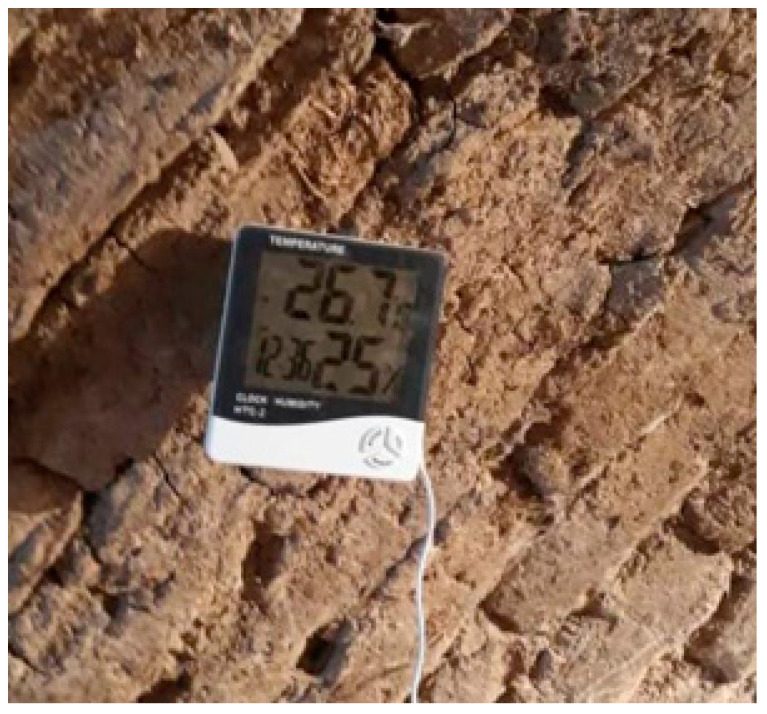
Photo of the digital temperature humidity meter.

**Figure 4 animals-13-02348-f004:**
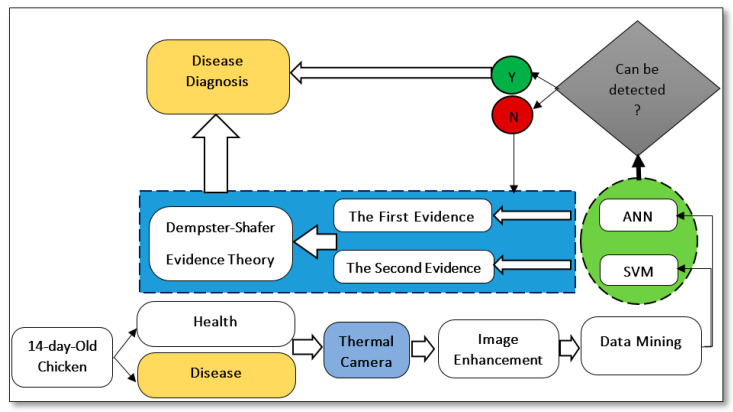
Proposed algorithm framework to identify avian diseases. ANN is artificial neural network, and SVM is support vector machine.

**Figure 5 animals-13-02348-f005:**
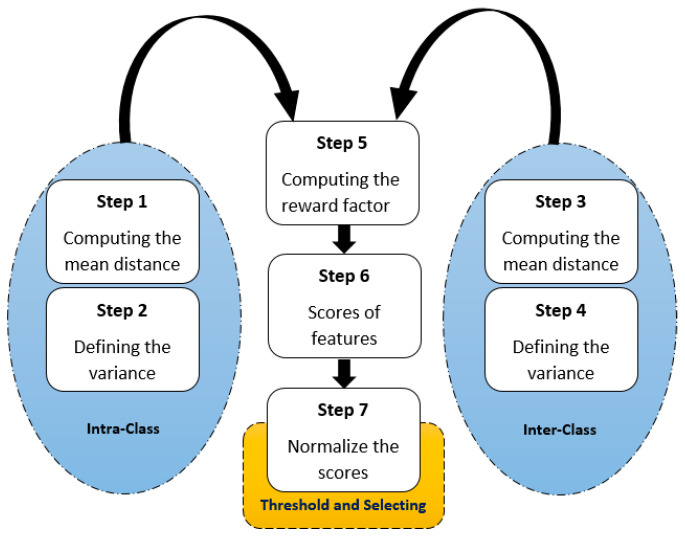
The seven steps of conducting the improved distance evaluation.

**Figure 6 animals-13-02348-f006:**
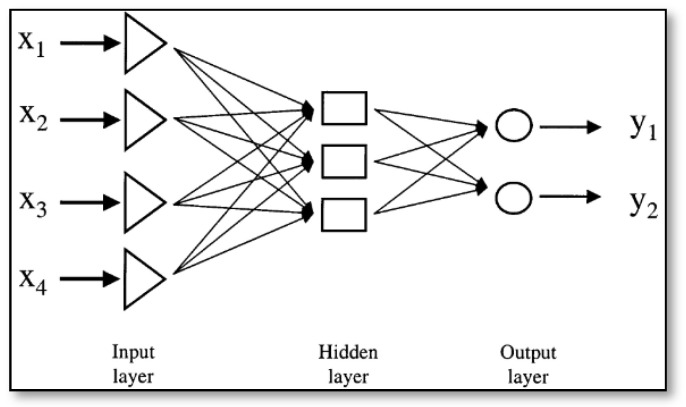
Architecture of the Artificial Neural Network.

**Figure 7 animals-13-02348-f007:**
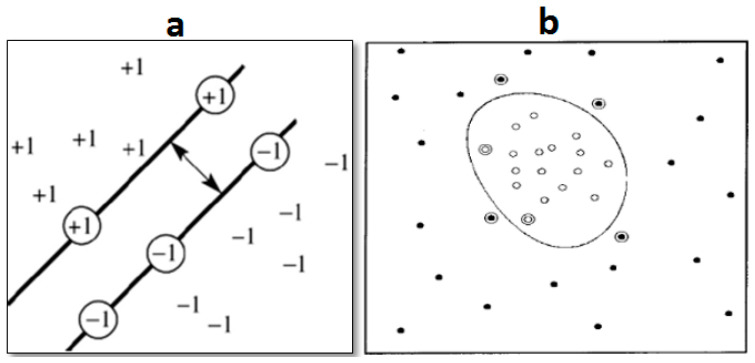
Concept of the hyperplanes in Support Vector Machine [32,33]: (**a**) linear hyperplane; (**b**) plane with Radial Basis Function.

**Figure 8 animals-13-02348-f008:**
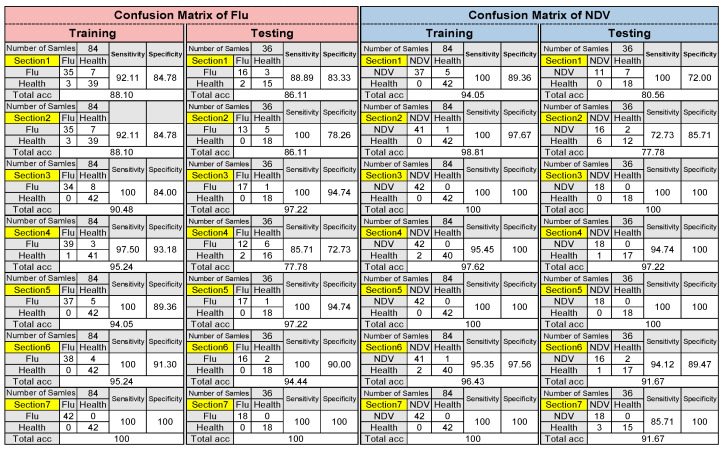
Confusion matrix of SVM for detecting Avian Influenza (Flu) and NDV (Newcastle Disease Virus) in all sections.

**Table 1 animals-13-02348-t001:** Formulas to extract features from chicken thermal images.

The Name of Feature	Formula for the Feature	The Name of Feature	Formula for the Feature
Mean	$F_{1}=\frac{\sum_{n=1}^{N}x(n)}{N}$	Geometric mean	$F_{13}=\sqrt[{N}]{\prod_{n=1}^{N}x(n)}$
Maximum	$F_{2}=\max\left|x(n)\right|$	Correlation coefficient	$F_{14}=\frac{F_{3}}{F_{2}}\times 100$
Standard deviation (std)	$F_{3}=\sqrt{\frac{\sum_{n=1}^{N}{\left(x\left(n\right)-F_{1}\right)}^{2}}{N-1}}$	The average deviation from the mean	$F_{15}=\sum_{n=1}^{N}\frac{\left(\left|x\left(n\right)\right|-F_{1}\right)}{N}$
Quadratic mean square	$F_{4}={\left(\frac{\sum_{n=1}^{N}\sqrt{\left|x(n)\right|}}{N}\right)}^{2}$	Skewness	$F_{16}=\frac{\sum_{n=1}^{N}{\left(x\left(n\right)-F_{1}\right)}^{3}}{F_{3}^{3}}$
Root mean square	$F_{5}=\sqrt{\frac{\sum_{n=1}^{N}{\left(x(n)\right)}^{2}}{N}}$	Kurtosis	$F_{17}=\frac{\sum_{n=1}^{N}{\left(x\left(n\right)-F_{1}\right)}^{3}}{(N-1){\left(F_{3}\right)}^{4}}$
Third central moment divided by the std	$F_{6}=\frac{F_{18}}{F_{3}^{3}}$	The third central moment	$F_{18}=\frac{1}{N}\sum_{n=1}^{N}{\left(x\left(n\right)-F_{2}\right)}^{3}$
Crest factor	$F_{7}=\frac{F_{2}}{F_{5}}$	The fourth central moment	$F_{19}=\frac{1}{N}\sum_{n=1}^{N}{\left(x\left(n\right)-F_{2}\right)}^{4}$
Maximum divided by the Quadratic mean square	$F_{8}=\frac{F_{2}}{F_{4}}$	The fifth central moment	$F_{20}=\frac{1}{N}\sum_{n=1}^{N}{\left(x\left(n\right)-F_{2}\right)}^{5}$
Root mean square divided by the mean	$F_{9}=\frac{F_{5}}{F_{1}}$	The sixth central moment	$F_{21}=\frac{1}{N}\sum_{n=1}^{N}{\left(x\left(n\right)-F_{2}\right)}^{6}$
Impulse factor	$F_{10}=\frac{F_{2}}{\frac{1}{N}\sum_{n=1}^{N}\left|x(n)\right|}$	The fourth central moment divided by the square of the variance	$F_{22}=\frac{F_{19}}{{\left(F_{11}\right)}^{2}}$
Variance	$F_{11}=\frac{\sum_{n=1}^{N}{\left(x\left(n\right)-F_{1}\right)}^{2}}{N-1}$	The sum of squares	$F_{23}=\sum_{n=1}^{N}{\left(x(n)\right)}^{2}$
Harmonic mean	$F_{12}=\frac{N}{\sum_{n=1}^{N}\frac{1}{x(n)}}$		

**Table 2 animals-13-02348-t002:** Definitions for True Positive (TP), False Positive (FP), True Negative (TN), False Negative (FN), and actual condition and predicted condition.

		Actual Condition	
		Unhealthy	Healthy
Predicted condition	Unhealthy	TP	FP
	Healthy	FN	TN

**Table 3 animals-13-02348-t003:** Feature scores of Group A (AI infection group + corresponding control group).

Features	Section 1	Features	Section 2	Features	Section 3	Features	Section 4	Features	Section 5	Features	Section 6	Features	Section 7
**F21**	**1.000**	**F22**	**1.000**	**F21**	**1.000**	**F12**	**1.000**	**F12**	**1.000**	**F22**	**1.000**	**F22**	**1.000**
**F2**	**0.835**	**F21**	**0.938**	**F2**	**0.882**	**F22**	**0.959**	**F14**	**0.848**	**F12**	**0.979**	**F12**	**0.904**
**F4**	**0.835**	**F4**	**0.925**	**F4**	**0.882**	**F14**	**0.864**	**F22**	**0.838**	**F2**	**0.915**	**F2**	**0.817**
**F22**	**0.764**	**F2**	**0.925**	**F22**	**0.881**	**F7**	**0.774**	**F7**	**0.832**	**F4**	**0.915**	**F4**	**0.817**
**F9**	**0.761**	**F12**	**0.874**	**F9**	**0.755**	**F8**	**0.774**	**F8**	**0.832**	**F14**	**0.910**	**F14**	**0.803**
F23	0.597	**F14**	**0.830**	F12	0.680	**F6**	**0.751**	**F6**	**0.804**	**F21**	**0.889**	**F19**	**0.795**
F1	0.497	**F9**	**0.758**	F3	0.677	**F15**	**0.751**	**F20**	**0.804**	**F7**	**0.851**	**F21**	**0.761**
F3	0.497	**F3**	**0.702**	F14	0.646	**F20**	**0.751**	**F15**	**0.804**	**F8**	**0.851**	**F7**	**0.740**
F16	0.494	F6	0.584	F23	0.638	**F2**	**0.703**	**F5**	**0.753**	**F6**	**0.822**	F18	0.598
F17	0.491	F15	0.584	F7	0.497	**F4**	**0.703**	F2	0.656	**F15**	**0.822**	F10	0.597
F12	0.472	F20	0.584	F8	0.497	F21	0.578	F4	0.656	**F20**	**0.822**	F6	0.590
F14	0.455	F7	0.562	F16	0.494	F10	0.431	F23	0.654	**F5**	**0.812**	F15	0.590
F18	0.385	F8	0.562	F5	0.479	F5	0.489	F10	0.408	**F9**	**0.771**	F20	0.590
F19	0.323	F5	0.525	F6	0.477	F9	0.468	F11	0.330	**F3**	**0.756**	F13	0.583
F11	0.363	F23	0.576	F20	0.477	F23	0.456	F21	0.521	F11	0.456	F9	0.549
F10	0.343	F11	0.320	F15	0.477	F11	0.381	F3	0.490	F10	0.451	F3	0.467
F7	0.258	F10	0.309	F11	0.308	F3	0.385	F9	0.477	F16	0.443	F8	0.459
F8	0.258	F16	0.439	F10	0.298	F1	0.365	F19	0.304	F13	0.282	F1	0.447
F5	0.242	F17	0.251	F17	0.370	F19	0.203	F18	0.206	F23	0.266	F11	0.590
F6	0.232	F1	0.089	F1	0.363	F18	0.157	F1	0.168	F17	0.236	F5	0.422
F15	0.232	F18	0.081	F18	0.232	F16	0.106	F16	0.166	F19	0.091	F17	0.312
F20	0.232	F13	0.056	F19	0.141	F13	0.058	F17	0.021	F1	0.070	F23	0.032
F13	0.078	F19	0.019	F13	0.120	F17	0.038	F13	0.017	F18	0.024	F16	0.008

Note: The bold font indicates selected features for each section.

**Table 4 animals-13-02348-t004:** Feature score for Group B (ND infection group + corresponding control group).

Features	Section 1	Features	Section 2	Features	Section 3	Features	Section 4	Features	Section 5	Features	Section 6	Features	Section 7
**F16**	**1.000**	**F16**	**0.987**	**F16**	**1.000**	**F17**	**1.000**	**F13**	**1.000**	**F16**	**1.000**	**F16**	**1.000**
**F9**	**1.000**	**F9**	**0.943**	**F2**	**0.985**	**F16**	**0.976**	**F2**	**0.943**	**F3**	**0.987**	**F9**	**0.939**
**F3**	**0.988**	**F17**	**0.845**	**F9**	**0.965**	**F2**	**0.943**	**F9**	**0.871**	**F9**	**0.947**	**F3**	**0.906**
**F17**	**0.929**	**F3**	**0.813**	**F3**	**0.897**	**F9**	**0.880**	**F16**	**0.869**	**F17**	**0.911**	**F17**	**0.815**
**F21**	**0.858**	**F5**	**0.805**	**F17**	**0.888**	**F3**	**0.878**	**F3**	**0.845**	**F2**	**0.891**	**F2**	**0.809**
**F4**	**0.816**	**F2**	**0.803**	**F19**	**0.812**	**F19**	**0.868**	F23	0.543	F5	0.699	F1	0.687
**F1**	**0.816**	F6	0.683	F18	0.684	F23	0.774	F17	0.427	F8	0.674	F4	0.687
**F2**	**0.810**	F15	0.683	F5	0.523	F21	0.757	F12	0.356	F7	0.674	F14	0.655
**F5**	**0.802**	F20	0.683	F8	0.513	F4	0.720	F22	0.337	F21	0.643	F5	0.636
F6	0.654	F1	0.662	F7	0.513	F1	0.720	F19	0.335	F6	0.629	F15	0.635
F15	0.654	F4	0.662	F6	0.509	F5	0.696	F18	0.310	F20	0.629	F6	0.635
F20	0.654	F21	0.660	F15	0.509	F8	0.679	F11	0.450	F15	0.629	F20	0.635
F7	0.645	F8	0.652	F20	0.509	F7	0.679	F14	0.232	F19	0.623	F7	0.605
F8	0.645	F7	0.652	F23	0.457	F6	0.643	F21	0.198	F14	0.616	F8	0.605
F22	0.636	F14	0.620	F11	0.450	F15	0.643	F7	0.191	F1	0.610	F21	0.583
F19	0.617	F19	0.620	F13	0.393	F20	0.643	F8	0.191	F4	0.610	F22	0.519
F14	0.615	F22	0.618	F14	0.371	F14	0.581	F15	0.169	F23	0.546	F12	0.503
F12	0.572	F12	0.618	F1	0.318	F22	0.560	F20	0.169	F22	0.513	F19	0.462
F23	0.542	F23	0.467	F4	0.318	F12	0.506	F6	0.169	F12	0.480	F11	0.450
F11	0.450	F18	0.461	F12	0.310	F11	0.450	F1	0.156	F11	0.450	F23	0.434
F13	0.309	F11	0.450	F10	0.218	F13	0.270	F10	0.156	F13	0.261	F13	0.375
F10	0.284	F10	0.134	F22	0.214	F10	0.197	F4	0.156	F10	0.243	F18	0.268
F18	0.272	F13	0.095	F21	0.180	F18	0.196	F5	0.146	F18	0.239	F10	0.209

Note: The bold font indicates selected features for each section.

**Table 5 animals-13-02348-t005:** The number of thermal images for each group.

Group	Classifier	Number of Thermal Images	Training	Validation	Testing
A or B	ANN	120	84	18	18
	SVM	120	84	0	36

Note: ANN is artificial neural network; SVM is support vector machine.

**Table 6 animals-13-02348-t006:** Detection of Avian Influenza using thermal images and ANN.

Disease	Classifier	The Time for Data Collection	Validation Accuracy (%)	Testing Specificity	Testing Sensitivity (%)	Training Accuracy (%)	Testing Accuracy (%)
Avian Influenza (AI or Flu)	ANN Structure: 5 × 8 × 2	Section 1	75.93	93.33	41.67	74.60	70.37
		Section 2	72.22	71.43	69.23	81.35	70.37
		Section 3	92.59	100.00	69.44	84.52	79.63
		Section 4	98.15	100.00	85.71	93.65	92.59
		Section 5	94.44	100.00	85.71	96.83	92.59
		Section 6	98.15	100.00	90.63	94.44	94.44
		Section 7	100.00	100.00	100.00	100.00	100.00

Note: The highlighted section indicates the earlier time for successful disease detection.

**Table 7 animals-13-02348-t007:** Detection of Newcastle Disease using thermal images and ANN.

Disease	Classifier	The Time for Data Collection	Validation Accuracy (%)	Testing Specificity (%)	Testing Sensitivity (%)	Training Accuracy (%)	Testing Accuracy (%)
Newcastle Disease (ND)	ANN Structure: 5 × 7 × 2	Section 1	68.52	77.78	59.26	74.21	68.52
		Section 2	87.04	72.73	75.00	78.57	74.07
		Section 3	74.07	86.96	77.42	83.33	81.48
		Section 4	62.96	64.29	53.85	69.05	59.26
		Section 5	100.00	100.00	100.00	100.00	100.00
		Section 6	92.59	85.71	63.16	89.68	77.78
		Section 7	92.59	83.33	83.33	87.30	83.33

Note: The highlighted section indicates the earlier time for successful disease detection.

**Table 8 animals-13-02348-t008:** SVM core parameters for the prediction of Avian Influenza (AI) and Newcastle Disease (ND).

Group	Kernel Function	σ	Solver	Box Constraint (C)
AI	Radial Basic Function	1	Sequential Minimal Optimization	5
ND				10

**Table 9 animals-13-02348-t009:** Detection of Avian Influenza using thermal images and SVM.

Disease	Classifier	The Time for Data Collection	Validation Accuracy (%)	Testing Specificity (%)	Testing Sensitivity (%)	Training Accuracy (%)	Testing Accuracy (%)
Avian Influenza (AI or Flu)	SVM (sigma = 1 & C = 5)	Section 1	86.11	83.33	88.89	88.10	86.11
		Section 2	86.11	78.26	100.00	88.10	86.11
		Section 3	97.22	94.74	100.00	90.48	97.22
		Section 4	77.78	72.73	85.71	95.24	77.78
		Section 5	97.22	94.74	100.00	94.05	97.22
		Section 6	94.44	90.00	100.00	95.24	94.44
		Section 7	100.00	100.00	100.00	100.00	100.00

Note: The highlighted section indicates the earlier time for successful disease detection.

**Table 10 animals-13-02348-t010:** Detection of Newcastle Disease using thermal images and SVM.

Disease	Classifier	The Time for Data Collection	Validation Accuracy (%)	Testing Specificity (%)	Testing Sensitivity (%)	Training Accuracy (%)	Testing Accuracy (%)
Newcastle Disease (ND)	SVM (sigma = 1 & C = 10)	Section 1	80.56	72.00	100.00	94.05	80.56
		Section 2	77.78	85.71	72.73	98.81	77.78
		Section 3	100.00	100.00	100.00	100.00	100.00
		Section 4	97.22	100.00	94.74	97.62	97.22
		Section 5	100.00	100.00	100.00	100.00	100.00
		Section 6	91.67	89.47	94.12	96.43	91.67
		Section 7	91.67	100.00	85.71	100.00	91.67

Note: The highlighted section indicates the earlier time for successful disease detection.

**Table 11 animals-13-02348-t011:** The results of data fusion for identifying Avian Influenza in Section 4 with Dempster–Shafer evidence theory.

Section 4		Infection	Healthy	Testing Sensitivity (%)	Testing Specificity (%)	Testing Accuracy (%)
Bird States	AI infection	100.00	0.00	98.15	100.00	99.05
	Health	1.89	98.11			

**Table 12 animals-13-02348-t012:** The results of data fusion for identifying Newcastle Disease in Section 2 with Dempster–Shafer evidence theory.

Section 2		Infection	Healthy	Testing Sensitivity (%)	Testing Specificity (%)	Testing Accuracy (%)
Bird States	ND infection	96.97	3.03	82.90	96.35	88.48
	Healthy	20.00	80.00			

**Table 13 animals-13-02348-t013:** Summary of the SVM performance to diagnose AI and ND using thermal images.

The Time for Data Collection	AI (Flu)			ND		
	Sensitivity (%)	Specificity (%)	Testing Accuracy (%)	Sensitivity (%)	Specificity (%)	Testing Accuracy (%)
Section 1	88.89	83.33	86.11	100.00	72.00	80.56
Section 2	100.00	78.26	86.11	82.90	96.35	88.48
Section 3	100.00	94.74	97.22	100.00	100.00	100.00
Section 4	98.15	100.00	99.05	94.74	100.00	97.22
Section 5	100.00	94.74	97.22	100.00	100.00	100.00
Section 6	100.00	90.00	94.44	94.12	89.47	91.67
Section 7	100.00	100.00	100.00	85.71	100.00	91.67

## Data Availability

Data is contained within the article.

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
