# Peer review of "Early Detection of Avian Diseases Based on Thermography and Artificial Intelligence"

_animals, 2023, doi:10.3390/ani13142348_

Round 1
Reviewer 1 Report
In this work, Sadeghi et al. investigated the thermographic image + artificial intelligence workflow to assist in the early detection of avian diseases. This work timely and properly contributes to the domain of precision livestock farming. The manuscript was finished well and from my perspective, I do not see any major changes required. However, there are several points that need further clarification. My comments are as follows.
Lines 95-98: was the intent to combine thermographic images and machine learning? There is existing literature that indicated the importance of thermography on poultry disease/welfare diagnostics. For instance,
Nääs, Irenilza Alencar, Rodrigo Garófallo Garcia, and Fabiana Ribeiro Caldara. "Infrared thermal image for assessing animal health and welfare." Journal of Animal Behaviour and Biometeorology 2.3 (2020): 66-72.
González, C., Pardo, R., Fariña, J., Valdés, M. D., Rodríguez-Andina, J. J., & Portela, M. (2017, October). Real-time monitoring of poultry activity in breeding farms. In IECON 2017-43rd Annual Conference of the IEEE Industrial Electronics Society (pp. 3574-3579). IEEE.
The authors may clearly define the gap and meanwhile, point out what has been done with thermography applications in avian/poultry science.
Material and methods: ANN or other deep learning models are well known to process raw data. Why was ANN applied after feature extraction rather than using thermographic images as the input?
Lines 334-335: SVM did better when the input was the features extracted from raw data. But that does not mean SVM outperforms ANN in any classification tasks.
Author Response
Dear Sir,
I have attached the referee comment's

Reviewer 2 Report
The authors have conducted an infection trial on broilers with Avian Influenza and Newcastle Disease, respectively, collected thermal image data of healthy and infected broilers and trained and evaluated two classifiers (ANN, SVM) for early detection of the two avian diseases based on features of the collected image data. The research is a first step towards automatic early detection of avian diseases on chicken farms, where an early treatment could prevent spreading of the disease, thus improve animal welfare and reduce economic losses.
My main concern is with the sample size, which is relatively small for machine learning applications (84 data points for training the models, respectively). Besides, broiler chickens where kept under experimental conditions in separate compartments and thermal images were taken manually at 7 different times and from a very close distance (50 cm). Hence, applicability to practical conditions with large numbers of chickens kept in one pen seems difficult.
Specific comments:
l. 66: It is stated that sound recognition is a popular automatic monitoring tool, but only one relatively old reference is stated [10]. Please add some more recent references for automatic sound detection to support this argument.
l. 75: Note that cameras installed in stables also gather dust and dirt.
Sec. 2.1.: Please clarify how many images were taken at each section. In l. 130 it says: “A total number of 240 thermal images…”. Does that mean 240 thermal images per section (= 840 images in total)? How many where taken from healthy, how many from unhealthy birds (60 each for AI and ND, respectively, at each section)? Where 3 images taken from each bird in each section or was this unevenly distributed? Were all taken images used in the next steps or was there a pre-selection?
l. 144: Please state (shortly) which methods were used for image enhancement.
Sec. 2.2., 2.3., 2.4: Which software/packages were used for feature extraction/selection and machine learning?
l. 156: “25 statistical features” -> “23 statistical features”;
l. 156: I would recommend changing “intensity for n data points” to “intensity for data point n”
Table 1: F8: Change to “Maximum divided by quadratic mean square” and maybe write F8 = F2 / F4; F17: For consistency, std could be replaced by F3.
l. 205 and Equations 1-6: The defined evaluation metrices are not in accordance with the ones used in Sec. 3 (e.g. True and False Positive Rates are not stated in Sec. 3). Please either delete the unused metrics here or add to the results section!
Table 2: Please add “actual condition” and “predicted condition” to the confusion matrix.
l. 222, l. 231: Please give details on if the thresholds (0.7 for Group A and 0.8 for Group B) were selected manually or automatically. For Group A, the difference between the scores for Section 3 and Section 5 does not look significant (same for Group B, Section 4).
l. 242: Please specify which cross validation strategy was used.
Table 6 & 7: Are Specificity and Sensitivity given for the overall data set (Training and Testing)?
l. 281, l. 291: I would recommend pointing out here that D-S evidence theory is applied in Sec. 3.3.
Table 11: I like the representation with confusion matrices, but accuracy values are already given in Tables 9 and 10.
l. 317: When was accuracy considered acceptable? According to Sec. 3.2, D-S is only used if SVM cannot detect with acceptable accuracy.
Table 14: Redundant, as the results can be found in Tables 9, 10, 12 and 13.
Author Response
I have attached the answers to the questions.
